# Epstein Barr Virus Hepatitis—A Mild Clinical Symptom or a Threat?

**DOI:** 10.3390/vaccines11061119

**Published:** 2023-06-19

**Authors:** Magdalena Rutkowska, Maria Pokorska-Śpiewak

**Affiliations:** Department of Children’s Infectious Diseases, Medical University of Warsaw, Regional Hospital of Infectious Diseases, 01-201 Warsaw, Poland; mpspiewak@gmail.com

**Keywords:** Epstein-Barr virus, alanine aminotransferase, aspartate aminotransferase, children, adolescents, viral hepatitis

## Abstract

The present study aimed to characterize pediatric patients diagnosed with hepatitis associated with primary Epstein-Barr Virus (EBV) infection. We described the changes in liver aminotransferases activity during the disease, and we analyzed the results of abdominal ultrasonography. A retrospective study was performed by analyzing the medical records of 166 immunocompetent children diagnosed with primary EBV hepatitis hospitalized at the Department of Children’s Infectious Diseases, Medical University of Warsaw, Regional Hospital of Infectious Diseases in Warsaw, between August 2017 and March 2023. Elevated alanine aminotransferase (ALT) activity was noted in the first three weeks of the disease. In 46.3% of patients, ALT values exceeded five times the upper limit of the laboratory norm in the first week of illness. Aspartate aminotransferase activity increased from the first to fourth week from the onset of symptoms and showed two peaks in the first and third weeks. The changes over time of mean AST activity demonstrated significance. Transient cholestatic liver disease was the predominant type of hepatic involvement in 10.8% of children; 66.6% of them were older than 15 years. Clinical and ultrasound criteria of acute acalculous cholecystitis (AAC) were met in three female patients over 16 years of age. Hepatitis associated with primary EBV infection is usually a mild and self-limiting condition. Significantly elevated values of liver enzymes with features of cholestatic liver disease may occur in patients with a more severe course of the infection.

## 1. Introduction

The Epstein-Barr virus (EBV, HHV-4, human gamma-herpesvirus 4) is a double-stranded DNA virus. EBV causes approximately 90% of infectious mononucleosis (IM), so-called glandular fever, which typically manifests itself as the classic triad of symptoms: fever, pharyngitis, and lymphadenopathy. The remaining 10% of IM cases are caused by: cytomegalovirus (CMV), human herpes virus 6, herpes simplex virus, and human immunodeficiency virus (HIV) [1]. EBV infection can take various clinical forms, as the virus can affect virtually any organ. EBV is an etiologic agent of other disorders as well, for example, Burkitt lymphoma, nasopharyngeal carcinoma, undifferentiated B- or T-lymphocyte lymphomas, leiomyosarcoma, X-linked lymphoproliferative syndrome, and post-transplantation lymphoproliferative disorders. EBV is known to cause 5.6% of infection-associated cancers [2]. EBV infection is estimated to be associated with the development of cancer in approximately 200,000 patients worldwide each year. Therefore, work is underway on an EBV vaccine [3,4].

The only known reservoir of EBV are humans, and over 95% of the worldwide adult population have serological features of past infection [5,6]. Transmission occurs through saliva (kissing, coughing, sharing of food), genital secretions, blood transfusion, and transplantation [6]. The incubation period lasts from 30 to 50 days [6]. It is known that high contagiousness of EBV may occur up to 180 days after the onset of symptoms [7]. Young children and teenagers are most often affected.

Usually, the course of IM is self-limiting with an excellent prognosis. In approximately 80–90% of cases, the infection has mild or no symptoms, particularly among young children [6,8]. The older the child is, the greater the risk of full-blown IM with fever, tonsillitis, lymphadenopathy, fatigue, and hepatosplenomegaly [9]. The symptoms last for about two to four weeks.

According to the literature review (most of the data relate to adult patients), primary EBV infection is in 80–90% of cases associated with hepatic involvement, which is characterized by a mild to moderate acute elevation of liver aminotransferases as an effect of hepatocellular injury [8]. These abnormalities often persist unrecognized. The underlying pathogenesis and immune mechanisms remain unclear. No standard diagnostic criteria or management recommendations are available. The severity of the condition varies from asymptomatic, self-limited icteric hepatitis to acute liver injury (in rare cases). Patients diagnosed with hepatitis due to primary EBV infection may develop cholestatic features. Adult patients present with jaundice more often than children. Hemolysis or cholestasis are potential causes of jaundice in IM. The condition is usually benign and resolves spontaneously within a few weeks [9]. However, little information is available about children and adolescents with hepatitis due to primary EBV infection.

The aim of the study was to characterize pediatric patients diagnosed with hepatitis associated with primary EBV IM. Our study was conducted on a fairly large cohort of patients. Most studies to date concern much smaller groups. The assumption of this work was to create a general characteristic of the study population. We described the changes in the activity of liver aminotransferases during the infection, and we analyzed the results of abdominal ultrasonography evaluation.

## 2. Patients and Methods

### 2.1. Patients

We retrospectively analyzed the data collected on patients aged 0–18 years managed for IM due to primary EBV infection at the Department of Children’s Infectious Diseases, Medical University of Warsaw, Regional Hospital of Infectious Diseases in Warsaw, between August 2017 and March 2023.

The onset of the disease was defined as the first day of fever, lymphadenopathy, sore throat, gastrointestinal symptoms, or jaundice. To confirm the primary EBV infection, we performed a serological test detecting immunoglobulin M antibody to EBV viral capsid antigen: anti-EBV VCA IgM (chemiluminescence immunoassay—CLIA test system, LIAISON EBV IgM, DiaSorin S.p.A., Saluggia, Italy) as a marker, which does not occur in chronic disease. The diagnosis of hepatitis related to primary EBV infection was made based on the increase of serum alanine aminotransferase (ALT) activity over age reference values [10].

We also performed more laboratory tests to assess liver function: serum aspartate aminotransferase and γ-glutamyl transferase (γ-GT) activity and bilirubin concentration with fractions. The laboratory investigation also included a full blood count with differential and C-reactive protein (CRP). We defined leukocytosis according to age and sex standards [11].

Abdominal ultrasonography was ordered in patients with a more severe course of the disease. Splenomegaly was diagnosed when the longitudinal dimension of the spleen exceeded the 97.5th percentile for age [12].

The following exclusion criteria were established:(1)CMV co-infection (positive CMV IgM),(2)pre-existing hepatitis or other underlying diseases with impaired liver function,(3)positive tests result for other viruses (hepatitis A virus—HAV, hepatitis B virus—HBV, hepatitis C virus—HCV) causing similar symptoms,(4)pre-existing medication, which could impair liver function.

### 2.2. Statistical Analysis

We analyzed collected medical records using STATISTICA 13.3 (StatSoft, Kraków, Poland). Data were presented as mean (Standard Deviation; SD) unless otherwise indicated. The chi-square (χ^2^) test was used to verify the normality of the variables’ distribution. Since none of the tested variables showed a normal distribution, the differences over time were compared through the non-parametric Kruskal-Wallis test for multiple comparisons. A two-tailed *p*-value of <0.05 was determined to be statistically significant.

## 3. Results

### 3.1. Demographic and Clinical Characteristics

The data of 415 pediatric patients hospitalized due to IM syndrome within 79 months were retrospectively analyzed. In this group, 199 (47.9%) patients developed hepatitis with elevated ALT activity. Of these patients, 33 (16.6%) were excluded from the study because of coinfection with CMV (n = 27) and potentially hepatotoxic medication due to underlying chronic disease (n = 6) (Figure 1).

One hundred sixty-six patients, aged 18 months to 18 years (mean, 11.9 ± 5.1 years), were enrolled in the study. Sixty-eight (40.9%) were male. There is an increased incidence of IM in two age groups: from 4 to 10 years (46 patients) and between 14 and 18 years of age (80 children). The age distribution is shown in Figure 2.

Forty-two (25.3%) patients suffered from other (chronic) diseases. Allergic conditions (bronchial asthma, allergic rhinitis, atopic dermatitis, and inhalant or food allergy) were the most common and concerned 23 (13.8%) patients. The second most common chronic disease was adenoid hyperplasia which occurred in five (3.0%) children.

### 3.2. Clinical Course and Complications

We analyzed the frequency of symptoms and signs in the study population. The average duration of symptoms was 14.4 ± 5.3 days and ranged from 4 to 30 days. The mean duration of hospitalization was 4.7 ± 2.2 days and ranged from 1 to 14 days.

The most common sign was lymphadenopathy, which was observed in 154 (92.8%) patients. Most often, the cervical lymph nodes were involved. Pharyngitis and tonsillitis with tonsillar exudates were the second most common clinical features and occurred in 150 (90.3%) patients. One hundred forty-three (86.1%) patients developed fever with an average duration of 8.0 ± 4.2 days (from 2 to 21 days). The classic triad of IM symptoms, including fever, lymphadenopathy, and pharyngitis or tonsillitis, was seen in 126 (75.9%) children.

Hepatomegaly and splenomegaly were noted in 134 (80.7%) and 111 (66.9%) patients, respectively. The concomitant enlargement of the liver and spleen presented 106 (63.8%) children. The degree of liver enlargement on palpation ranged from 0.5 to 10 cm (mean, 2.5 ± 1.3 cm) below the costal arch. Seventy-seven (46.4%) patients suffered from gastrointestinal symptoms such as abdominal pain and nausea or vomiting. A rash was seen in 55 (33.1%) children; among them, 28 (50.9%) developed an allergic-toxic generalized rash after previous amoxicillin administration. The frequency of signs and symptoms is presented in Table 1.

Concomitant infections were recognized in eighty-three patients (50.0%). The most common diagnosis was bacterial pharyngitis (based only on the clinical picture or confirmed by microbiological testing). Four patients were diagnosed with cervical lymphadenitis (2.4%), and three (1.8%) children developed sepsis. The incidence of the accompanying infections is shown in Table 2.

Eighty patients (48.2%) developed complications not related to the liver, biliary tract, or gallbladder, which occurred before or during hospitalization. The obturation of the upper respiratory tract due to severe nasopharyngeal and palatal tonsils enlargement was the most common and was observed in 69 (41.5%) children. In four (2.4%) patients, the EBV infection was complicated with thrombocytopenia without purpura. Table 3 presents the incidence of particular complications.

None of the study patients developed fulminant hepatitis, and no cases of EBV-hepatitis-related mortality were noted.

### 3.3. Laboratory and Imaging Test Results

The leucocyte count was evaluated in all patients (166) with values ranging from 4470 to 49,000 cells/µL (mean 14,621 ± 6888). Leukocytosis with lymphocytosis was observed in 129 patients (77.7%); thirty-three (19.9%) children had normal white blood cells count with relative lymphocytosis. The median lymphocyte percentage in white blood cell smear was 67.7 ± 13.6%, and atypical lymphocytes: 25.6 ± 14.5%. The lowest number of platelets was 9800, and the highest—was 455,000 (mean 209,823 ± 66,780).

We assessed inflammation markers: C-reactive protein (CRP)—in 161 patients and procalcitonin (PCT)—only in children with a more severe course of the disease. CRP ranged from normal values (<5 mg/dL) to 75 mg/dL. PCT serum concentration ranged from 0.05 to 37.0 ng/mL.

The activity of lactate dehydrogenase (LDH) in serum was assessed only in 20 patients with a mean value of 686.95 ± 232.25 U/L. Table 4 shows laboratory values on admission.

The activity of both enzymes ALT and AST were measured once or more in 164 (98.8%), twice or more in 83 (50.0%) and three or more times in eight (4.8%) children. Table 5 presents the number and percentages of patients with different values of hepatic laboratory parameters in the course of the disease.

An average of 1.71 (±0.84) evaluation of AST activity was performed in 164 (98.8%) patients before or during hospitalization. The changes over time of mean AST activity showed statistical significance (*p* = 0.0124). Elevated AST levels were observed even in the first week of the illness and persisted until the fourth week showing two peaks: in the first (mean 186.34 ± 148.17 IU/L) and third (mean 169.00 ± 140.98 IU/L) week. A decreasing tendency was noted from the fourth week, with normalization in the fifth week from the onset of the disease.

ALT was measured on average 1.79 (±0.86) times for each child. ALT levels showed similar variability over time as AST, but there were no peaks of activity observed—ALT remains at a comparable elevated level for the first three weeks of the disease (mean 234.16 ± 164.77 IU/L).

γ-GT evaluation was performed from one to four times (mean 1.64 ± 0.72) in thirty-six patients. Mean γ-GT activity was above the upper normal limit in the first 3 weeks of the disease onset, reaching the highest values in the second week (mean 183.84 ± 97.48 IU/L).

Total serum bilirubin concentration was evaluated from one to three times (mean 1.26 ± 0.55) in thirty-eight patients; in twenty-three of them (60.5%), it exceeded the upper limit of the laboratory norm. Mean total bilirubin was elevated in the first two weeks of the disease (mean 2.0 ± 2.00 mg/dL), and normalization was observed from the third week.

In 18 (47.4%) patients, conjugated bilirubin was at least 20% of total bilirubin, all of them had elevated activity of γ-GT and prothrombin ratio in normal ranges from 72 to 123% (mean 88.9 ± 9.6%). In this group (10.8% of the study population), transient cholestatic liver disease was the predominant type of hepatic involvement. All of them presented with jaundice, pruritus, and abdominal tenderness; eight (44.4%) children reported darkening of the urine and two (11.1%)—acholic stools. The mean age in the group of patients with cholestatic features was 13 ± 5.23 years, and 12 (66.6%) children were older than 15 years (Figure 3).

ALT activity and total serum bilirubin concentration exceeding five times the upper limit of the laboratory norm were noted mainly in the first week of the disease in 46.3% and 13% of patients, respectively (ALT level and total serum bilirubin concentration were evaluated in the first week of the disease in 95 and 23 patients, respectively). AST and γ-GT activity more than five times the upper limit of the laboratory norm occurred predominantly in the second week of illness and occurred in 30.6% and 70.8% of children, respectively (the above parameters were measured in the second week of the disease in 108 and 24 patients, respectively). Changes in ALT, AST, γ-GT, and total bilirubin levels during the disease are shown in Figure 4.

The abdominal ultrasound was performed in 116 (69.8%) patients, and abnormalities were observed in 111 (66.8%) cases. Splenomegaly was the most common cause of abnormal results and occurred in 99 (85.3%) children. Hepatomegaly was shown in 72 (43.4%) children, and enlargement of the lymph nodes of the hepatic hilum—in 64 (38.5%) cases. In 97 (83.6%) out of 116 patients who underwent abdominal ultrasonography, an enlarged liver was found on palpation. This finding was confirmed by abdominal ultrasound in 64 (65.9%) children. Eight patients (6.9%) were diagnosed with hepatomegaly by ultrasonography, which was not detected by palpation. Abnormalities related to the gallbladder and biliary tract were shown in 19 (11.4%) patients, and most often occurred the gallbladder wall thickening—in 17 (10.2%) cases. Gallbladder polyps and fluid collection in the hepatic hilum were observed in two (1.2%) and three (1.8%) patients, respectively.

### 3.4. Treatment

All patients required supportive care due to high fever, dehydration, upper respiratory tract obturation, or severe allergic-toxic rash after amoxicillin.

Ninety-one (54.8%) patients received an empirical antimicrobial therapy before the hospitalization (and before the establishment of the diagnosis), most often due to pharyngitis. Among them, 23 (25.6%) were treated with several antibiotics: 24 (26.4%) received two antibiotics, and 4 (4.4%) received three. Forty-three (47.2%) children were prescribed amoxicillin, and 28 (65.1%) of them developed an allergic-toxic rash 7–10 days after the first dose of the medication. The number (and percentage) of patients treated with particular antibiotics are presented in Table 6.

During the hospitalization, 85 (51.2%) children required antibiotic therapy. The most common indication for the antimicrobial regimen was concomitant bacterial pharyngitis. Five patients (5.8%) received two antibiotics. Table 7 shows the number of patients treated with particular antibiotics during the hospitalization.

In 27 (29.7%) of the ninety-one patients previously treated with antibiotics, the therapy was continued using the same medication, and in 25 (27.5%) patients, the therapy was changed. The antibiotic therapy was discontinued in 39 (42.8%) children because of its ineffectiveness or occurrence of side effects (for example, allergic-toxic rash after amoxicillin). Thirty-three (19.8%) patients required initiation of antibiotic therapy.

## 4. Discussion

In our retrospective study, we characterized a group of 166 pediatric patients diagnosed with EBV-hepatitis. The age distribution in the examined population indicates the dominance of two age groups: 4–8 years and 14–18 years. In the first group of 41 patients (24.7%), there is a slight female predominance (53.6%). This trend was more pronounced in the second group of 90 patients aged 14–18, where girls constitute 62.2%.

The pathogenesis of EBV-hepatitis remains unclear. Infection of hepatocytes with primary hepatotropic viruses, such as hepatitis B or C, has no direct cytopathic effect, and symptoms of liver injury result from an immune response to viral antigens presented by infected hepatocytes [13,14]. Hepatocytes, biliary and sinusoidal epithelium are not the target cells for EBV [15]. The possible pathogenetic mechanism of parenchymal injury of the liver tissue and cholestasis in EBV infection involves the effect of the virus on the systemic and intrahepatic synthesis of pro-inflammatory cytokines that affect the sinusoidal and tubular bile transport systems. EBV-infected CD8+ T cells in the liver are known to secrete inflammatory mediators such as tumor necrosis factor α, interferon-γ, and Fas ligand. Another explanation could be an inhibition of antioxidant mechanisms through the production of autoantibodies [15,16].

Available literature data in children indicate that 80–90% of patients with primary EBV infection have clinical and/or laboratory evidence of hepatitis [8,17,18]. Our study demonstrated that liver involvement during acute EBV infection develops in approximately 50% of hospitalized children, which is inconsistent with previous reports. In our study cohort, hepatitis was mild and self-limited, and cholestatic features were present in 10.8% of patients with hepatitis. It is known that cholestatic hepatitis in primary EBV infection occurs more commonly with increasing age and has been reported mainly in adults [19]. In our study, 66.6% of patients with clinical and laboratory features of cholestasis were older than 15 years, which is consistent with previous reports.

Forty (24.1%) patients did not present the classic triad of IM symptoms (fever, lymphadenopathy, and pharyngitis or tonsillitis).

The abnormalities of ALT and AST serum activity in our study were observed from the first to the third week of the disease and started returning to normal in the fourth week. Our findings are consistent with the existing literature [8,20]. Values of the above parameters exceeding five times the upper limit of the laboratory norm were most often observed in the first week of the disease for ALT (46.3% of patients) and in the second week for AST (30.6%). These results are, in part, inconsistent with studies in the adult population, which state that serum aminotransferase levels are usually elevated by less than five times the upper normal limit [17,18,21].

Our study demonstrated two peaks in AST levels: in the first and third week of illness. A similar pattern of changes in AST activity over time was described in 41 adult patients by Kofteridis et al.

Abdominal ultrasonography in EBV infection may reveal splenomegaly, hepatomegaly, porta hepatis adenopathy, periportal edema, and fluid collection or gallbladder wall thickening (GBWT). A gallbladder wall thickness greater than 3 mm is considered abnormal [22]. The last of the listed abnormalities is recognized as a sign of the severity of hepatitis. A possible explanation could be an immune reaction (as seen in hepatitis) or swelling of the gallbladder wall as a result of lymphatic obstruction due to the enlargement of porta hepatis lymph nodes [23]. Dehydration as a result of pharyngitis and poor appetite can also increase bile viscosity.

GBWT greater than 3 mm in the absence of cholelithiasis in patients with abdominal pain localized in the right upper quadrant is defined as acute acalculous cholecystitis (AAC). Such gallbladder inflammation accounts for approximately 30–50% of cholecystitis in the pediatric population [24]. Abdominal ultrasonography in AAC may reveal biliary sludge, gallbladder distention (hydrops), or pericholecystic fluid as well. AAC as a complication of viral infection in children is rare but was reported in primary CMV infection, EBV infection, and human herpes virus type 6 infection [25,26,27,28,29].

In our study, 18 patients presented ultrasonographic signs of AAC, but only three of them reported abdominal pain localized in the right upper quadrant. These three patients were females older than 16 years. It is consistent with previous reports, which describe female predominance in AAC in both adult and pediatric populations with EBV infection [27]. AAC, as a complication of EBV infection, is usually associated with a favorable prognosis [30].

Immunocompetent patients with symptomatic EBV infection usually spontaneously recover with supportive care (antipyretics, analgesics, hydration, and rest). In our study, children with complications in the form of upper respiratory tract obturation received systemic corticosteroids. Antibiotics were administered in cases of concomitant signs and symptoms of bacterial infection, especially in the presence of significantly elevated inflammatory markers such as CRP and PCT or microbiological confirmation. All our patients achieved full recovery without any chronic complications.

In conclusion, although our study has some limitations, namely its retrospective nature and unavailability of all laboratory data, it demonstrates that primary EBV hepatitis is mostly a mild and transient condition. No case of acute liver failure was reported in this study group. In the available literature, isolated cases of fatal liver failure—even in immunocompetent patients—have been reported [31]. Therefore, it is important to determine the activity of liver enzymes and monitor them in patients with a more severe course of IM. Delayed presentation of patients to the physician may have a negative impact on our results. Not all children had performed additional tests at the beginning of the disease. We analyzed the course of hepatitis only in children requiring hospitalization due to primary EBV infection.

## Figures and Tables

**Figure 1 vaccines-11-01119-f001:**
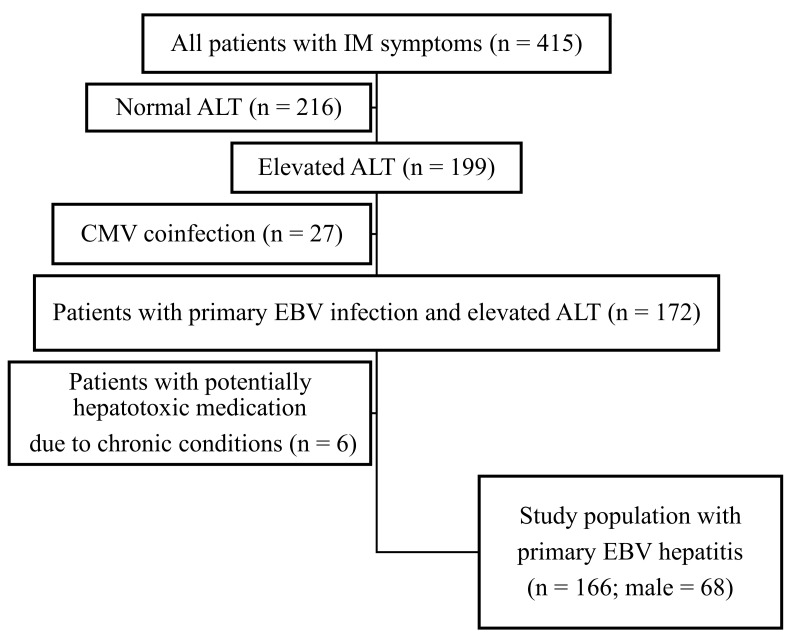
Flowchart showing the enrollment of patients. EBV, Epstein-Barr virus; ALT, alanine aminotransferase; CMV, cytomegalovirus.

**Figure 2 vaccines-11-01119-f002:**
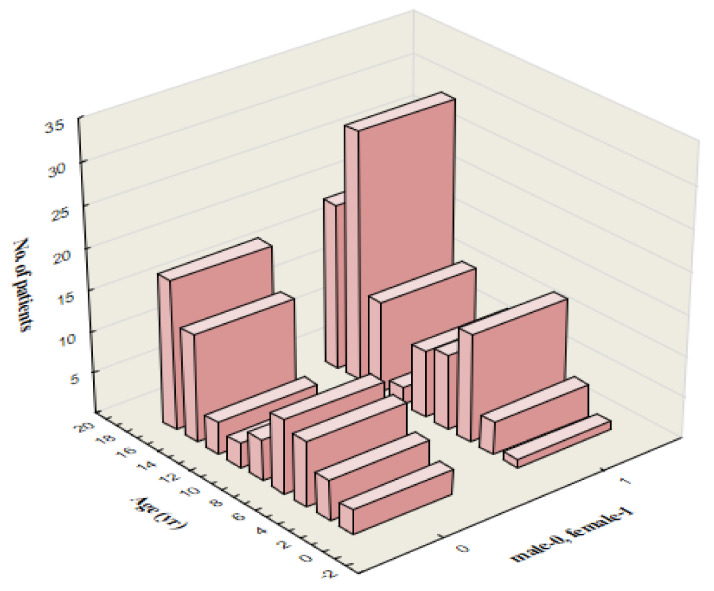
Age distribution in the study population (166 patients) by sex of the patients (male = 68). EBV, Epstein-Barr virus.

**Figure 3 vaccines-11-01119-f003:**
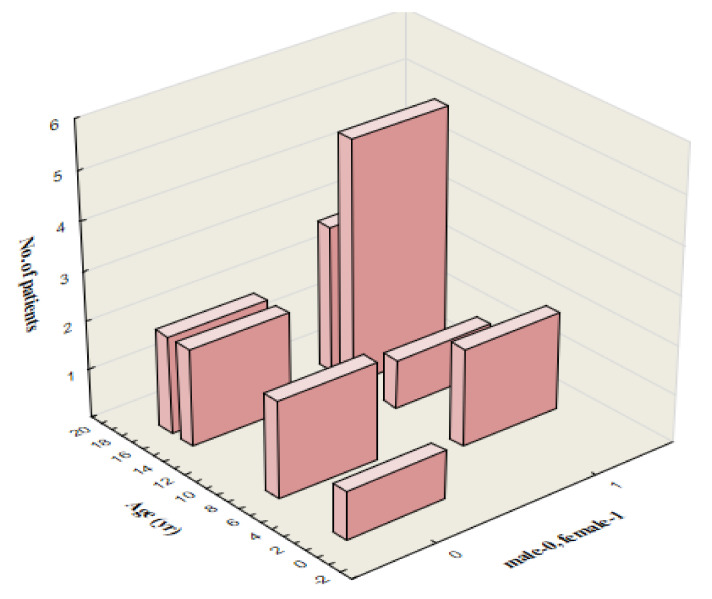
Age distribution by sex of 18 patients with cholestatic liver disease in primary EBV hepatitis (male = 7). EBV, Epstein-Barr virus.

**Figure 4 vaccines-11-01119-f004:**
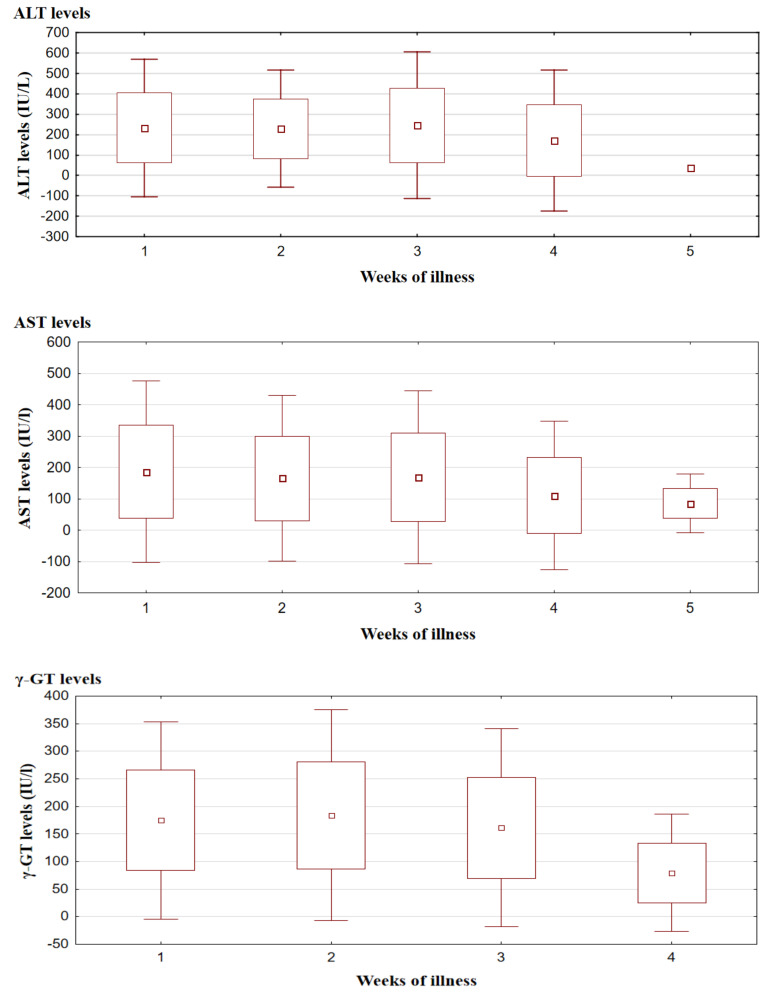
Mean activity of ALT, AST, γ-GT, and total serum bilirubin concentration (+/− standard deviation) during EBV infection. Kruskal-Wallis test was used for changes over time: *p* = 0.0124 for AST. No significant changes over time for ALT (*p* = 0.0581), γ-GT (*p* = 0.0988), and total serum bilirubin (*p* = 0.8562). ALT—alanine-aminotransferase; AST—aspartate-aminotransferase; γ-GT—γ-glutamyl transferase.

**Table 1 vaccines-11-01119-t001:** Frequency of signs and symptoms in 166 patients with primary EBV hepatitis.

Signs and Symptoms	No. of Patients (%)
Lymphadenopathy	154 (92.8)
Pharyngitis with tonsillar exudate	150 (90.3)
Fever	143 (86.1)
Hepatomegaly	134 (80.7)
Splenomegaly	111 (66.9)
Rash	55 (33.1)
Abdominal pain	52 (31.3)
Eyelid swelling	49 (27.7)
Nausea or vomiting	45 (27.1)
Rash after amoxicillin administration	28 (16.9)
Jaundice/icteric sclera	19 (11.4)
Pruritus	19 (11.4)
Abdominal tenderness	18 (10.8)
Dark urine	8 (4.8)
Acholic stools	2 (1.2)

**Table 2 vaccines-11-01119-t002:** The incidence of concomitant infections in 166 patients with primary EBV hepatitis.

Infection	No. of Patients (%)
Bacterial pharyngitis	51 (30.7)
Acute otitis media	8 (4.8)
Oral candidiasis	7 (4.2)
Scarlet fever	5 (3.0)
Pneumonia	4 (2.4)
Lymphadenitis	4 (2.4)
COVID-19	3 (1.8)
Sepsis	3 (1.8)
Urinary tract infection	3 (1.8)
Herpes labialis	2 (1.2)
Herpes zoster	1 (0.6)
Varicella	1 (0.6)
Influenza	1 (0.6)

**Table 3 vaccines-11-01119-t003:** The incidence of particular complications other than the liver, biliary tract, or gallbladder pathology.

Complication	No. of Patients (%)
Obturation of the upper respiratory tract	69 (41.5)
Thrombocytopenia with or without purpura	4 (2.4)
Urticaria	3 (1.8)
Epistaxis	2 (1.2)
Syncope/loss of consciousness	2 (1.2)
Chest pain	2 (1.2)
Anemia	1 (0.6)
Iatrogenic inflammation and venous thrombosis	1 (0.6)
Vertigo	1 (0.6)

**Table 4 vaccines-11-01119-t004:** Laboratory values on the admission of the 166 patients with EBV-hepatitis.

Laboratory Value	Mean (±SD)	No. of Patients
White blood cells count (cells/µL)	14,621 (6,8)	166
Neutrophils (%)	31.9 (13.2)	162
Lymphocytes (%)	67.7 (13.6)	162
Atypical lymphocytes (%)	25.6 (14.5)	162
Platelets (cells/µL)	209,823 (66,7)	166
CRP (mg/dL)	2.8 (16.4)	161
PCT (ng/mL)	1.92 (6.9)	28
LDH (U/L)	686.95 (232.2)	20

**Table 5 vaccines-11-01119-t005:** Number and percentages (%) of patients with different values of ALT, AST, γ-GT, and total bilirubin during EBV infection.

		Number of Patients (%)			
		ALT	AST	γ-GT	Total Bilirubin
1st week ^a^	Normal	0 (0)	0 (0)	0 (0)	12 (52.2)
	<2×	12 (12.6)	24 (25.5)	1 (5.6)	4 (17.4)
	2–5×	39 (41.1)	42 (44.7)	5 (27.8)	4 (17.4)
	>5×	44 (46.3)	28 (29.8)	12 (66.6)	3 (13.0)
2nd week ^b^	Normal	3 (2.6)	7 (6.5)	0 (0)	9 (50.0)
	<2×	19 (16.2)	30 (27.7)	2 (8.4)	2 (11.1)
	2–5×	53 (4.3)	38 (35.2)	5 (20.8)	5 (27.8)
	>5×	42 (35.9)	33 (30.6)	17 (70.8)	2 (11.1)
3rd week ^c^	Normal	1 (1.8)	2 (4.2)	0 (0)	4 (66.6)
	<2×	7 (13.0)	15 (31.2)	0 (0)	2 (33.4)
	2–5×	24 (44.5)	19 (39.6)	3 (27.3)	0 (0)
	>5×	22 (40.7)	12 (25.0)	8 (72.7)	0 (0)
4th week ^d^	Normal	2 (10.0)	3 (18.7)	0 (0)	2 (66.6)
	<2×	6 (30.0)	7 (43.8)	2 (33.4)	1 (33.4)
	2–5×	9 (45.0)	4 (25.0)	3 (50.0)	0 (0)
	>5×	3 (15.0)	2 (12.5)	1 (16.6)	0 (0)
>4 weeks ^e^	Normal	1 (50.0)	0 (0)	0 (0)	0 (0)
	<2×	0 (0)	1 (33.4)	0 (0)	0 (0)
	2–5×	0 (0)	2 (66.6)	0 (0)	0 (0)
	>5×	1 (50.0)	0 (0)	0 (0)	0 (0)

^a^ ALT was evaluated in 95 patients, AST was evaluated in 94 patients, γ-GT was evaluated in 18 patients, and total bilirubin was evaluated in 23 patients; ^b^ ALT was evaluated in 117 patients, AST was evaluated in 108 patients, γ-GT was evaluated in 24 patients, and total bilirubin was evaluated in 18 patients; ^c^ ALT was evaluated in 54 patients, AST was evaluated in 48 patients, γ-GT was evaluated in 11 patients, and total bilirubin was evaluated in 6 patients; ^d^ ALT was evaluated in 20 patients, AST was evaluated in 16 patients, γ-GT was evaluated in 6 patients, and total bilirubin was evaluated in 3 patients; ^e^ ALT was evaluated in 2 patients, AST was evaluated in 3 patients, γ-GT was evaluated in 0 patients, and total bilirubin was evaluated in 0 patients.

**Table 6 vaccines-11-01119-t006:** The number (and percentage from ninety-one cases) of patients treated with particular antibiotics before hospitalization.

Antibiotic	No. of Patients (%)
Cefuroxime axetil	31 (34.1)
Amoxicillin with clavulanic acid	29 (31.8)
Phenoxymethylpenicillin	16 (17.6)
Azithromycin	15 (16.5)
Amoxicillin	14 (15.4)
Cefadroxil	6 (6.6)
Clarithromycin	5 (5.5)
Trimethoprim/Sulfamethoxazole	1 (1.1)
Cefazoline	1 (1.1)
No data on a specific preparation	2 (2.2)

**Table 7 vaccines-11-01119-t007:** The number (and percentage from eighty-five cases) of patients treated with particular antibiotics during the hospitalization.

Antibiotic	No. of Patients (%)
Cefuroxime or Cefuroxime axetil	66 (77.6)
Clindamycin	8 (9.4)
Clarithromycin	6 (7.0)
Penicillin G or Phenoxymethylpenicillin	4 (4.7)
Ceftriaxone	4 (4.7)
Azithromycin	2 (2.3)
Metronidazole	2 (2.3)

## Data Availability

The data presented in this study are available on request from the corresponding author. The data are not publicly available, because the study based on data collected in the Department’s database.

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
