# Peer review of "Epstein Barr Virus Hepatitis—A Mild Clinical Symptom or a Threat?"

_vaccines, 2023, doi:10.3390/vaccines11061119_

Round 1
Reviewer 1 Report
Rutkowska et al. retrospectively studied EBV-hepatitis in 166 immunocompetent children. They found that hepatitis by primary EBV usually shows a mild and self-limiting feature. These features of HBV-hepatitis were known well and no novel findings were observed.
Major
1. Mild clinical features of EBV-hepatitis are generally known well and no novel findings were found in this manuscript. Authors should show new findings (Foe example; an association between the grade of hepatitis and clinical features or laboratory data).
Minor
1. Authors should carefully use the term “cholestatic liver disease” because cholestatic liver injury may not occur in all patients.
2. “DiaSorin” should not be bold (line 74).
3. There are several mistypes (lines 154, 172, 256, 263).
Thickening of the gallbladder wall during viral hepatitis is known well as the result of contagion of liver inflammation, not the inflammation of the gallbladder itself. The description in lines 316 – 323 may give wrong information to the readers.
Minor English editing is required.
Reviewer 2 Report
Estimated Authors,
I've read with great interest the present article reporting on EBV and hepatic disorders in a sample of around 400 patients (with 166 final observations). The paper is interesting, but several improvements are required. More precisely:
Please include in "introduction" section a more extensive reporting on EBV hepatitis, as the introduction in its current stage of development over represents EBV in general over EBV hepatitis (the main topic of the study).
Statistical analysis is in fact non-existent, while even a very basic univariate/bivariate analysis would radically improve the quality of this study (see further).
Please revise Figure 2 by including the gender of participants.
Sentence "We assessed the frequency of symptoms and signs in the study population" is redundant and could be omitted.
Please be consistent in reporting your results, if in row 120 you did firstly report average and then range, follow this blueprint also in row 121.
Because of the design of this study (i.e. EBV hepatitis) it could be of some interest evaluate whether hepatomegaly and/or jaundice were associated or not with other signs/symptoms (e.g. through a very basic chi squared test). The very same for Table 2 and Table 3.
I've noticed a certain inconsistence in reporting decimal figures; please double check, choice 2 or 3 decimal figure and please stick with this option (at least, within the very same group of variable).
Figure 3, the very same for Figure 2.
Conclusions include a section that would more properly fit "limits" section. Moreover, Authors should address another limitation: while they have initially recruited well over 400 cases, the final sample is well under 200, that is a more than 50% dropout! Authors should address whether this reduction from the initial sample could have affected the representivity of the study or not, and whether the choice for having focused on patients with abnormal ALT could have impaired the reliability of your results (as ALT levels are particularly high in the first 3 weeks of the disease, could have a late access to medical service led to the improper exclusion of cases that did have hepatomegaly and simply obtained an early improvement?).
In fact, the paper is largely acceptable. I warmly recommend a double check but nothing more.
Round 2
Reviewer 1 Report
1. I understand this study uses a larger cohort of patients, compared with previous studies. So, the authors should describe this in the introduction part to clarify the aim of this study.
2. Various font sizes of characters are seen. Please revise them.
Reviewer 2 Report
Estimated Authors,
I've appreciated the considerable efforts to cope with my previous recommendations. Unfortunatley, I've noticed that figures have not been edited according to my recommendations. Currently, Figures 2 and 4 will required a minor adjustment in order to include the actual sample size of the various age group calculated. Thefore, I'm recommending the potential acceptance od the paper after the aforementioned managing.
The overall quality of the English text is substantially acceptable.
